# Sarcomatoid and Aggressive Variants in High R.E.N.A.L. Score T1b RCC: Outcomes After Laparoscopic and Robotic Radical Nephrectomy

**DOI:** 10.3390/cancers17243918

**Published:** 2025-12-07

**Authors:** Murad Asali, Galeb Asali, Ron Batash, Moshe Schaffer

**Affiliations:** 1Urology Department, Barzilai Medical Center, Ben Gurion University of the Negev, Beer Sheva 8430905, Israel; 2Assuta Medical Center, Beer Sheva, Ramat Hyal, Ben Gurion University of the Negev, Beer Sheva 8489507, Israel; 3Faculty of Medicine in Safed, Bar-Ilan University, Ramat-Gan 1311502, Israel; galeb.asali@live.biu.ac.il; 4Orthopedic Oncology, Orthopedic Department, Wolfson Medical Center, Tel-Aviv University, Holon 5822012, Israel; ronb@wmc.gov.il; 5Oncologic Department, Barzilai Medical Center, Ben Gurion University of the Negev, Beer Sheva 7830604, Israel; moshesc@bmc.gov.il

**Keywords:** renal cell carcinoma, T1b RCC, radical nephrectomy, laparoscopic surgery, R.E.N.A.L. nephrometry score, renal function preservation

## Abstract

Kidney cancer is typically managed with surgical intervention; nevertheless, when tumors are sizable or situated in challenging locations, therapeutic options become more intricate. Surgeons often aim to excise only a portion of the kidney to maintain R.E.N.A.L. function; however, this may not be feasible when tumors exhibit complex morphology or positioning, as indicated by the R.E.N.A.L. score. Certain cancers exhibit aggressive characteristics, particularly uncommon variants such as sarcomatoid transformations, which are associated with poorer prognoses despite the malignancy being diminutive or contained. This study examines individuals with moderate-sized kidney tumors (T1b) and elevated R.E.N.A.L. scores who underwent total nephrectomy via minimally invasive methods. The objective is to enhance comprehension of surgical outcomes, renal function, and the existence of aggressive tumor classifications. These findings may enhance treatment options and assist surgeons in reconciling cancer management with long-term health.

## 1. Introduction

RCC is the most common type of kidney cancer in adults, accounting for approximately 90–95% of cases [1]. It demonstrates a male predominance with a male-to-female ratio of up to 2:1 [2] and is most commonly diagnosed in elderly patients, particularly those over 75 years old [3].

The staging of RCC is crucial for determining prognosis and guiding treatment strategies. According to the American Joint Committee on Cancer (AJCC) 8th Edition Cancer Staging Manual, T1b tumors are defined as those confined to the kidney, measuring more than 4 cm but not more than 7 cm in greatest dimension [4].

The R.E.N.A.L. nephrometry score, introduced by Kutikov and Uzzo (2009), has become a standardized system for assessing the anatomical complexity of renal tumors [5]. Higher scores have been associated with more aggressive tumor pathology and increased surgical complexity [6]. This scoring system has proven valuable in preoperative planning and predicting postoperative outcomes [7].

Recent developments in artificial intelligence have shown promise in automating R.E.N.A.L. scoring, as demonstrated by Heller et al., who found that AI-generated scores were comparable to human-generated scores in predicting outcomes [8]. However, the clinical decision-making process continues to be complex, especially for tumors with high R.E.N.A.L. scores. The management of T1b RCC typically involves surgical intervention, with the main options being partial nephrectomy, radical nephrectomy, or, in select cases, active surveillance. Partial nephrectomy is often preferred when feasible, as it allows for preservation of renal function while achieving oncological control. Radical nephrectomy remains an appropriate choice in cases where tumor complexity or anatomical location precludes a nephron-sparing approach. Active surveillance may be considered in patients with significant comorbidities or limited life expectancy, where the risks of surgery outweigh the potential benefits. The choice of treatment is influenced by several factors, including tumor size, anatomical location, growth pattern, and proximity to critical structures such as blood vessels and the renal collecting system. Additionally, the patient’s overall health status, baseline renal function, and personal preferences play a key role in treatment planning. Recent advancements in surgical techniques, particularly the adoption of minimally invasive approaches such as laparoscopic and robotic-assisted surgery, have significantly improved perioperative and postoperative outcomes. These innovations have been associated with reduced blood loss, shorter hospital stays, faster recovery times, and comparable oncological results when compared to open surgery. As a result, patients with localized RCC now benefit from safer, more effective surgical options that not only aim for cancer control but also prioritize quality of life and renal function preservation and improved outcomes for patients with localized RCC [9].

While partial nephrectomy is increasingly advocated for T1b tumors [10], the optimal surgical approach for tumors with high R.E.N.A.L. scores remains debatable. Kim et al. suggested that partial nephrectomy could be considered for high R.E.N.A.L. score tumors but noted the need for caution due to potential recurrence risks [11]. This study intends to assess the results of laparoscopic radical nephrectomy in patients with T1b RCC and moderate-to-high R.E.N.A.L. scores, focusing on surgical outcomes, renal function preservation, and pathological findings.

Understanding the clinical presentation, staging, and management options for T1b RCC is essential for optimizing patient care and improving survival outcomes.

Jiang et al., in a review dealing with T1b RCC, concluded that PN may provide comparable outcomes in terms of RFS and OS and better renal function preservation, although CSS was worse [12].

## 2. Materials and Methods

### 2.1. Study Design and Patient Population

This retrospective study included patients who underwent laparoscopic radical nephrectomy for T1b RCC with R.E.N.A.L. scores ≥8 between 2008 and 2024. We chose a score of 8 and above because in all of the involved patients with T1b, there were no instances with a score lower than 8, so a score of 8 is considered moderate. Following the methodology established in previous studies [5,11], we collected and analyzed demographic data, perioperative outcomes, and pathological findings.

### 2.2. Surgical Technique

All procedures were performed using a standardized laparoscopic transperitoneal approach, following principles described in the current literature [9]. The technique included standard port placement, careful dissection around the renal hilum, and en bloc vessel stapling and excision of the kidney together with the tumor.

### 2.3. Follow-Up Protocol

Postoperative monitoring included regular assessment of renal function through creatinine measurements at day 1, one month, three months, six months, one year, and then biannually up to 30 months.

All patients return for follow-up one month after surgery with laboratory tests. An upper abdominal and urinary tract ultrasound is performed at three months, and an additional CT or ultrasound examination is recommended during the first postoperative year, then every six months for the first three years, and annually thereafter. Complications were recorded and classified according to standard criteria.

### 2.4. Statistical Analysis

Statistical analysis was performed using appropriate tests for continuous and categorical variables, following methodologies described in similar studies [10]. ANOVA and Kruskal–Wallis tests were used to analyze creatinine trends over time. The software we used for the statistical analysis was GraphPad Prism v10.4.0, released in 2024.

## 3. Results

### 3.1. Patient Demographics and Tumor Characteristics

The study included 118 patients (62 male; 56 female) with a mean age of 66.9 years and a mean BMI of 29.4. Right-sided tumors were present in 53 cases and left-sided in 65 cases. The average R.E.N.A.L. score was 10.02, with a mean tumor diameter of 6.0 cm by CT and 4.7 cm by pathology (Table 1). The mean follow-up was 8.22 years and the median follow-up duration was 8.35 years.

### 3.2. Perioperative Outcomes

The mean operative time was 151.6 min, with average blood loss of 75.2 mL. The mean hospital stay was 3.2 days. Nine complications (7.6%) were recorded: three incisional hernias, one deep vein thrombosis, one atrial fibrillation, one ileus, one splenic bleeding event, one wound infection, and one trocar site hematoma, all managed conservatively (Table 2).

### 3.3. Renal Function Outcomes

Creatinine levels showed an initial increase postoperatively but stabilized during follow-up, consistent with patterns observed in previous studies [12]. Mean creatinine values, both for individuals with and without pre-existing renal failure, are presented in Table 3.

### 3.4. Pathological Findings

Table 4 summarize the pathological findings, indicating aggressive pathological features in 35 cases (29.7%).

### 3.5. Survival

Table 5 presents survival outcomes for 118 patients over two follow-up periods: 2.5 years and 5 years after surgery.


**
*Short-Term Survival (2.5 Years)*
**


**Overall Survival (OS):** 100%**Cancer-Specific Survival (CSS):** 100%

No **deaths** occurred—either from any cause or cancer-related—within the first 2.5 years after surgery. These results reflect **excellent short-term outcomes.**


**
*Long-Term Survival (5 Years)*
**


**5-Year Overall Survival (OS):** 89.5%**5-Year Cancer-Specific Survival (CSS):** 97.4%

## 4. Discussion

In managing T1b renal cell carcinoma (RCC), both partial nephrectomy (PN) and radical nephrectomy (RN) are effective surgical alternatives, each possessing unique benefits and considerations.


*Radical Nephrectomy (RN):*

*Pros:*


**Oncological Control:** RN provides excellent oncological control by completely removing the affected kidney, which may reduce the risk of local recurrence [10].**Surgical Simplicity:** RN is generally less technically challenging than PN, potentially resulting in shorter operative times and lower immediate surgical risks.


*Cons:*


**Renal Function Decline:** Removal of the entire kidney can lead to a significant decline in renal function, increasing the risk of chronic kidney disease and its associated complications [12].**Long-Term Health Risks:** Studies have linked RN to higher rates of cardiovascular events and overall mortality, likely due to the resultant reduction in renal function [12].

The choice between PN and RN should be individualized, considering factors such as tumor size, location, patient comorbidities, and surgeon expertise. While PN offers the benefit of renal function preservation, RN may be more appropriate in cases where complete tumor excision is challenging with PN. Ongoing advancements in surgical techniques continue to refine the risk–benefit profile of both procedures in the management of T1b RCC.

Yang et al. concluded that patients with T1b RCC, OS, cancer-specific survival (CSS) and metastasis-free survival (MFS) are superior in patients receiving LPN than those treated with LRN. With the benefit of preserving renal function in LPN, which leads to lower incidence risk of other systematic diseases, LPN may be the preferred option when the condition permits for cases involving T1b RCC [10].

Recurrence rates following nephrectomy for renal cell carcinoma (RCC) are influenced by several factors, including tumor stage, surgical margins, and the type of nephrectomy performed.

In this study, PN patients were not included. Introducing a PN cohort would require a separate selection pathway, introduce non-equivalent anatomy, and risk further bias because PN was not routinely offered for R.E.N.A.L. ≥8 tumors during the study period. Including a small, highly selected PN subgroup would itself exacerbate selection bias.


*Overall Recurrence Rates:*


Approximately 20% of patients undergoing surgery for localized RCC experience recurrence post-nephrectomy. The likelihood of recurrence is higher in individuals with advanced tumor stages and grades at the time of surgery [13].


*Partial Nephrectomy (PN) vs. Radical Nephrectomy (RN):*


Studies comparing PN and RN have found no significant difference in recurrence-free survival between the two procedures. For instance, a systematic review and meta-analysis reported similar 5-year and 10-year recurrence-free survival rates for both PN and RN in patients with 4 to 7 cm RCCs [12].


*Positive Surgical Margins (PSMs):*


The presence of PSMs—where cancer cells are detected at the edge of the removed tissue—has been associated with an increased risk of recurrence, particularly after PN. A multi-institutional retrospective study indicated that PSMs following PN independently predicted a 75% increased risk of cancer recurrence [12].


*Tumor Complexity:*


Tumor complexity, frequently evaluated using the R.E.N.A.L. nephrometry score, can influence recurrence rates. Research suggests that patients with high-complexity tumors (R.E.N.A.L. score ≥ 10) undergoing PN have comparable 5-year recurrence-free survival rates to those undergoing RN. However, careful patient selection is crucial, as higher tumor complexity may be associated with increased recurrence risk [11].


*Risk Assessment Tools:*


Nomograms, such as the one developed by Memorial Sloan Kettering Cancer Center, are available to predict the probability of remaining recurrence-free survival five years post-nephrectomy. These tools consider factors like tumor size, stage, and histology to provide individualized risk assessments [14].

There are no articles dealing simultaneously with moderate-to-high R.E.N.A.L. tumor score in T1b, medium-term renal function follow-up, and aggressiveness.

Kim et al. concluded that PN, if technically feasible, could be considered to preserve renal function in patients with R.E.N.A.L. ≥10. Nonetheless, PN needs to be implemented with caution in some patients due to the higher potential for recurrence and poor survival. In their article, they did not discuss the aggressiveness of the tumor [11].

Kutikov et al. (2011) were the first to objectify the relationship between tumor anatomy and pathology [6]. Using the nephrometry score, they developed a tool to quantitate the preoperative likelihood of malignant and high-grade pathology of an enhancing renal mass.

Canter et al. showed in their article that R.E.N.A.L. nephrometry score standardizes the reporting of solid renal masses and appears to effectively stratify by treatment type. Although only one part of the treatment decision-making process, nephrometry aids in objectifying previously subjective measures [7].

Heller et al. presented automated AI-generated R.E.N.A.L. scores comparable to human-generated R.E.N.A.L. scores and predicted a wide variety of meaningful patient-centered outcomes [8]. This AI-based scoring was intended to facilitate wider adoption of the R.E.N.A.L. score.

In summary, while both PN and RN offer effective oncological control for RCC, factors such as surgical margins, tumor complexity, and individual patient characteristics play significant roles in determining recurrence rates. Close postoperative monitoring and tailored follow-up strategies are essential to detect and manage potential recurrences promptly.


*Aggressiveness in High R.E.N.A.L. Score RCC:*


**Tumor Grade Correlation:** Higher R.E.N.A.L. scores have been associated with more aggressive tumor pathology. A study evaluating the efficacy of the R.E.N.A.L. nephrometry score found that larger tumor sizes and higher R, E, and L scores were more likely to be associated with high-grade pathological outcomes [15].**Surgical Complexity:** Tumors with high R.E.N.A.L. scores (10–12) are considered highly complex, often posing significant challenges during nephron-sparing surgeries like partial nephrectomy. Such complexity may increase the risk of perioperative complications and influence the choice between partial and radical nephrectomy [16].**Predictive Value:** The R.E.N.A.L. nephrometry score serves as a valuable tool in preoperative planning, helping to predict the aggressiveness of renal masses and guiding the surgical approach. Higher scores may indicate a need for more extensive surgical intervention and vigilant postoperative monitoring due to the increased risk of aggressive tumor behavior [17].

In summary, a high R.E.N.A.L. nephrometry score in RCC is indicative of increased tumor aggressiveness and surgical complexity, necessitating careful preoperative assessment and tailored surgical strategies to optimize patient outcomes.

Our findings demonstrate that laparoscopic radical nephrectomy is a viable option for T1b RCC with a moderate-to-high R.E.N.A.L. nephrometry score. The relatively low complication rate (7.6%) and stable long-term renal function suggest that the procedure is safe and effective in this patient population, supporting previous findings by Yang et al. [10].

The high incidence of aggressive pathological features (29.7%) justifies the use of radical nephrectomy in these cases. This finding aligns with studies by Kutikov et al. [6] and Chen et al. [15] showing a correlation between high R.E.N.A.L. scores and aggressive tumor characteristics.

This study’s findings regarding the relationship between moderate-to-high R.E.N.A.L. scores and tumor aggressiveness support the work of Kutikov and Uzzo [5] and subsequent validation studies by Canter et al. [7]. The correlation between high R.E.N.A.L. scores and aggressive pathological features, as noted by Millet et al. [17], was clearly demonstrated in this cohort.

Sarcomatoid differentiation in RCC represents one of the most aggressive pathological features encountered across all tumor stages. Although rare, occurring in approximately 5–8% of RCCs, its presence is associated with rapid progression, early metastasis, and poor survival outcomes irrespective of tumor stage or histological subtype. In our cohort, sarcomatoid and angiosarcomatous variants were identified even within T1b tumors, emphasizing that tumor size alone does not fully capture aggressiveness. The fact that 29.7% of patients overall harbored aggressive features, including sarcomatoid variants, underscores the oncological risk posed by moderate-to-high R.E.N.A.L. score lesions, despite their “localized” staging.

Several studies have confirmed that sarcomatoid differentiation predicts poor outcomes even in apparently organ-confined RCC. Cheville et al. reported that sarcomatoid change is an independent predictor of decreased cancer-specific survival, with median survival often less than 2 years in advanced cases [18]. Mian et al. demonstrated that sarcomatoid features can occur even in small tumors and are strongly correlated with metastatic potential and resistance to conventional therapies [19]. These findings suggest that for patients with high R.E.N.A.L. score T1b tumors, the presence of aggressive variants may justify the selection of radical nephrectomy over nephron-sparing surgery, particularly when technical complexity also compromises margin control.

The sarcomatoid component shows malignant spindle cells that may grow as intersecting fascicles, which is reminiscent of pleomorphic undifferentiated sarcoma. Rhabdoid cells are epithelioid cells with eccentrically located vesicular nuclei with prominent nucleoli and large intracytoplasmic eosinophilic inclusions. Studies have shown that RCCs with sarcomatoid and rhabdoid differentiation have distinctive molecular features [20].

Sarcomatoid change in renal cell carcinoma portends a worse prognosis because tumors with even a small component of sarcomatoid change may have an adverse outcome, this finding, when present, should be noted in the surgical pathology report [21].

Moreover, the high R.E.N.A.L. score itself may indirectly function as a proxy for aggressiveness. Tumors with higher scores are frequently central, endophytic, and abutting the collecting system or renal hilum. These anatomical features not only complicate surgical excision but have been linked to a higher incidence of high-grade and sarcomatoid differentiation. Kutikov et al. initially demonstrated this correlation, and subsequent validation studies reinforced that R.E.N.A.L. score stratification may help predict which tumors harbor aggressive biology [6]. Our findings corroborate this notion: approximately one-third of tumors with R.E.N.A.L. ≥ 8 exhibited aggressive or uncommon histological variants, indicating that nephrometry-based preoperative planning should be regarded not merely as an indicator of surgical complexity but also as an indirect evaluation of oncological risk.

The clinical implication is that with high R.E.N.A.L. score T1b RCC, radical nephrectomy remains a reasonable option when sarcomatoid characteristics are suspected or confirmed, given the better oncological management provided by radical excision. This technique may exceed the long-term hazards of renal function decline in carefully chosen patients, particularly when weighed against the high mortality of aggressive histology. Future multicenter prospective studies examining the relationship between R.E.N.A.L. score, sarcomatoid differentiation, and survival outcomes will be required to improve treatment recommendations for this difficult subgroup.

The analysis of renal function demonstrates that despite initial postoperative elevation in creatinine levels, mid-term renal function remains stable in most patients. When excluding patients with pre-existing renal impairment (Cr > 1.2 mg/dL), the creatinine trends showed only mild fluctuations from POD1 through 30 months of follow-up (Figure 1), with final levels (1.16 mg/dL at 30 months) approaching baseline values (0.93 mg/dL preoperatively). This stability in renal function, particularly evident in patients without pre-existing renal impairment, suggests that the remaining kidney provides adequate functional compensation following radical nephrectomy, supporting findings from previous studies [10,12].

Comorbidities such as hypertension can worsen renal function after partial nephrectomy [22]. We do not have a control group, and this is an important aspect to investigate in the future in T1b renal cell carcinoma with moderate-to-high R.E.N.A.L. scores.

Although it is known that BMI worsens renal function after partial nephrectomy, we do not have a control group for comparison. However, our patients did have a relatively high BMI, and still, renal function did not show significant deterioration, as seen in our results [23]. We assume that renal function was preserved mainly for two reasons: first, due to the low percentage of patients with diabetes, and second, because of compensatory renal adaptation, particularly in patients who had normal preoperative renal function.


*Multifocality:*


Multifocal RCC is strongly associated with higher local recurrence rates and a higher probability of underlying aggressive biology.This is supported by multiple studies demonstrating that multifocality correlates with hereditary patterns, papillary subtype aggressiveness, and worse oncologic outcomes [24,25].

Recent SEER database analysis (Yang et al., 2024) demonstrated that patients with multiple tumors had significantly worse cancer-specific survival (hazard ratio ~1.43). Given our cohort’s high-complexity anatomy and >1 tumor count in 6.8%, we believe that the presence of multifocality represents a biologically more aggressive phenotype, warranting its inclusion as an aggressive pathological feature [25].


*Survival:*


The data in Table 5 demonstrate excellent cancer control, with minimal cancer-related mortality over five years. The slight drop in overall survival indicates that non-cancer-related factors may have contributed to long-term mortality. These findings support the efficacy and safety of the surgical approach used in this cohort.

While overall survival decreased by 5 years (from 100% to 89.5%), the cancer-specific survival remained very high at 97.4%. This suggests that most deaths beyond 2.5 years were due to non-cancer causes, highlighting the importance of considering patient comorbidities and age-related risks in long-term care.

The fact that only two patients died from cancer-related causes (as inferred from the CSS rate) further emphasizes the oncological effectiveness of the intervention.


*Study Limitations*


The study’s retrospective design and absence of a comparison group are significant drawbacks. Additionally, longer follow-up would be beneficial to assess very long-term outcomes, as suggested by previous studies [11].

## 5. Conclusions

Laparoscopic radical nephrectomy is a safe and efficient treatment for T1b renal cell carcinoma with moderate-to-high R.E.N.A.L. scores. The high incidence of aggressive pathological features in our cohort (29.7%) supports this surgical approach. Mid-term renal function remains stable in most patients despite initial postoperative elevation in creatinine levels. When excluding patients with pre-existing renal dysfunction, the creatinine trends showed only very minor change from POD1 and approached baseline values.

However, careful patient selection remains essential, especially for those with pre-existing renal impairment or other comorbidities that might affect long-term renal outcomes. Future prospective multi-center studies comparing different surgical approaches in this specific patient population would be valuable, with particular attention to long-term renal function outcomes beyond 30 months.

## Figures and Tables

**Figure 1 cancers-17-03918-f001:**
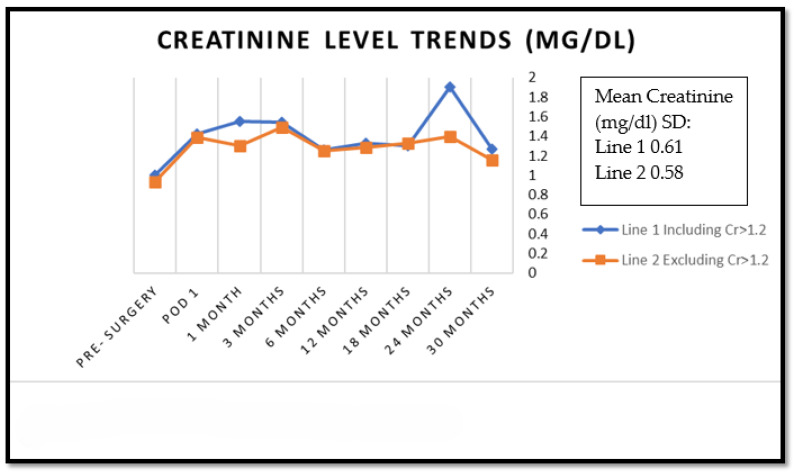
The figure illustrates the trends in mean creatinine (Cr) levels. The graph illustrates two distinct lines: the first encompasses all patients, including those with creatinine levels exceeding 1.2 mg/dL, while the second line excludes these patients. It can be observed that the graphs are almost overlapping at different time points and remain close to the baseline value.

**Table 1 cancers-17-03918-t001:** Characteristics.

Age (years) (Mean)	66.9
BMI (Mean)	29.4
Gender (M/F)	62/56
Laterality (R/L)	53/65
Mean R.E.N.A.L. Score	10.02
Tumor Diameter by CT (cm)	6
Tumor Diameter by Pathology (cm)	4.7
Diabetes Mellitus	4 (3.4%)
Hypertension	20 (16.9%)

**Table 2 cancers-17-03918-t002:** Perioperative findings and complications.

No.	118
Blood Loss (mL)	75.2
Operative Time (min)	151.6
Hospital Stay (days)	3.2
Complications	9
Incisional Hernia	3
Deep Vein Thrombosis	1
Atrial Fibrillation	1
Ileus	1
Spleen Bleeding	1
Wound Infection	1
Trocar Site Hematoma	1

**Table 3 cancers-17-03918-t003:** Mean creatinine (mg/dL) level trends summary including patients with renal failure.

Preoperative	POD 1	One Month	3 Months	6 Months	1 Year	*p* Value (ANOVA)
1	1.42	1.55	1.54	1.26	1.33	0.051
**	1.42	1.55	1.54	1.26	1.33	0.991
	18 months	24 months	30 months			
	1.3	1.9	1.27			0.989
Pre-op to 30 months post-op	**	**	**	**	**	0.59
POD 1 to 30 months	**	**	**	**	**	0.969
Creatinine Level Trends Summary, Excluding Patients with Renal Failure						
Pre-op	POD 1	One month	3 months	6 months	1 year	*p* value (ANOVA)
0.93	1.39	1.3	1.49	1.25	1.29	0.0074
**	1.39	1.3	1.49	1.25	1.29	0.993
	18 months	24 months	30 months			
	1.33	1.4	1.16			0.989
Pre-op to 30 months	**	**	**	**	**	0.433 Kruskal–Wallis Test
POD 1 to 30 months	**	**	**	**	**	0.428 Kruskal–Wallis Test

Pre-op: Pre-operation; POD—Postoperative Day. **: Indicates that the values are as listed above.

**Table 4 cancers-17-03918-t004:** Pathology results post nephrectomy—aggressiveness.

Grade 3	Multiple Tumors	Adrenal Involved	Lymph Node Involved	T3a	T3b	Total (%)
12	8	1	1	8	1	31 (26.3%)
Angiosarcoma	UCC					
2	2					4 (3.4%)
Total						35 (29.7%)
Angiomyolipoma	Oncocytoma					
2	4					6 (5.1%)

**Table 5 cancers-17-03918-t005:** Overall (OS) and Cancer-Specific Survival (CSS).

**Patients (n)**	**OS (2.5 Years)**	**CSS (2.5 Years)**	**5-Year OS from the Date of** **Surgery**	**5-Year CSS from the Date of Surgery**
100	100%	100%	68/76 (89.5%)	74/76 (97.4%)

## Data Availability

The data are not publicly available due to privacy and ethical restrictions, but are available from the corresponding author upon request.

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
