# Peer review of "Sarcomatoid and Aggressive Variants in High R.E.N.A.L. Score T1b RCC: Outcomes After Laparoscopic and Robotic Radical Nephrectomy"

_cancers, 2025, doi:10.3390/cancers17243918_

Round 1
Reviewer 1 Report
Comments and Suggestions for Authors
Overall Assessment
This study raises an interesting and important question regarding the significant proportion (approximately 30%) of cT1b renal cell carcinoma (RCC) cases that harbor aggressive pathological features. However, the current methodology, which exclusively focuses on patients who underwent Radical Nephrectomy (RN), introduces a significant selection bias that may limit the generalizability of the conclusions. It is plausible that the actual incidence of aggressive pathological features in the overall cT1b cohort (including those who underwent Partial Nephrectomy [PN]) might be lower. The authors' discussion hinges on a clear explanation of the criteria used to triage cT1b tumors for RN at their institution. Given this and other major methodological issues, I believe this manuscript requires Major Revision.
Major Comments
- Definition and Rationale for Aggressive Pathological Features
The inclusion of multiple tumors as an aggressive pathological feature lacks clear justification and supporting evidence in the current manuscript. The authors must provide the rationale for this classification, ideally with supporting literature.
The exclusion of pT3a (pathological upstaging) from the definition of "aggressive pathological features" is questionable, as upstaging to pT3a is a well-established prognostic factor for poor outcomes. The authors should present the reason for excluding pT3a or include it in their definition and analysis.
- Follow-up Protocol and Oncological Outcomes
The Materials and Methods section fail to detail the oncological follow-up protocol. The frequency of imaging modalities (e.g., CT scans, bone scintigraphy) must be clearly described.
The median follow-up duration is missing and should be reported.
The claim of reporting oncological outcomes at the 5-year mark appears highly premature, as the study cohort includes cases who underwent RN as recently as 2024. This late inclusion makes the long-term survival data unreliable and potentially misleading. The authors should either remove the discussion of 5-year oncological outcomes or exclude the recent cases to present a more robust and realistic analysis of long-term survival.
- Issues with Table 3 and Figure 1 (Renal Function Data)
Fundamental Patient Data is Missing: The presentation of renal function outcomes in Table 3 and Figure 1 is critically flawed due to the complete absence of baseline patient information. There is no presentation of the cohort's preoperative serum creatinine levels (other than the mean value mentioned in the Discussion) or the prevalence of key comorbidities (e.g., hypertension, diabetes mellitus) in the main results section. This makes it impossible to properly interpret the reported postoperative renal function trends.
Table 3: The data in the 4th row should arguably be bolded for emphasis, similar to the final row. More critically, the units of measurement for the values presented must be clearly stated. Since ANOVA was used, it is assumed the table presents the mean Cr values, but this needs explicit confirmation. The lack of detailed explanation is unacceptable.
Figure 1: The authors must clarify whether the Cr levels plotted in Figure 1 represent the mean values for all patients or only the subgroup without pre-existing renal impairment.
The figure should present more robust statistical descriptors than just the mean (e.g., quartiles or standard deviations/standard errors) to represent the distribution of the data accurately. These poor statistical presentation standards must be significantly revised.
The authors must address the specific trend observed: why did the Cr level for "all comers" deteriorate at 24 months and subsequently improve at 30 months? They should investigate whether this reflects a true trend or is due to the influence of outliers or sample size changes, and provide a clear explanation.
Minor Comment
Table 4: The 3rd row of Table 4 should arguably be presented in bold font for consistency and emphasis.
Author Response
Dear Editors,
Thank you very much for your valuable and thoughtful comments.
Here are our explanation and revisions:
- This article focus was on T1b disease strictly confined to the kidney by imaging criteria. However, as pathological upstaging (pT3a) reflects true extra-renal extension and carries poor prognosis. This pathological report is obtained after nephrectomy, and this is exactly what we aim to demonstrate — that sometimes the final pathology reveals a different and more aggressive histology or a more advanced stage that was not known preoperatively, we have added wording to the title of Table 4 to clarify that it refers to postoperative findings.
- We have added the sentence to the Patients and Methods section:All patients return for follow-up one month after surgery with laboratory tests. An upper abdominal and urinary tract ultrasound is performed at three months, an additional CT or ultrasound examination is recommended during the first postoperative year, then every six months for the first three years, and annually thereafter.
-
Mean follow-up: 8.22 years and the median follow-up duration is 8.35 years, we have added the calculations of the mean and median follow-up durations to the Results section.
- Regarding overall survival, we excluded the cases operated on in 2023–2024, leaving 100 patients for the calculation of 2.5-year overall survival. For the 5-year survival analysis, we excluded cases (Table 5) operated on less than five years ago, leaving 76 patients included in the calculation.
-
We have added to Table 1 the cases with diabetes mellitus and hypertension, including their percentages.
6. We have added the units of measurement for creatinine to Table 3.
7. We have changed the figure 1. title to “Mean creatinine level trends (mg/dL)”.
8. Figure 1. illustrates the trends in mean creatinine (Cr) levels. The graph illustrates two distinct lines: the first encompasses all patients, including those with creatinine levels exceeding 1.2 mg/dl, while the second line excludes these patients.
9. We assume that renal function was preserved mainly for two reasons:
first, due to the low percentage of patients with diabetes, and second, because of compensatory renal adaptation, particularly in patients who had normal preoperative renal function. We have added this note to the discussion section.10. We have made the change with the necessary words highlighted in Table 4.
With sincere appreciation and best regards,
Dr. Murad Asali
(on behalf of all co-authors)
Reviewer 2 Report
Comments and Suggestions for Authors
This study focuses on the outcomes of laparoscopic and robotic radical nephrectomy in patients with T1b renal cell carcinoma (RCC) and moderate to high RENAL scores (8 and above). The authors analyzed surgical outcomes, renal function, and pathological findings in a retrospective cohort of 118 patients. Among the rarer tumor pathologies, they identified angiosarcoma, urothelial carcinoma, angiomyolipoma, and oncocytoma. The manuscript includes five tables and one figure. I would recommend adding standard deviations (SD) to the creatinine level graph for better clarity. The manuscript cites 23 references, which is an appropriate number for a study of this scope. The authors report a high incidence of aggressive pathological features (29.7%) within their cohort. This finding supports the rationale for choosing radical over partial nephrectomy in T1b renal masses with RENAL scores of 8 or higher.
Author Response
Dear reviewer,
In response to Reviewer 2, who was highly impressed with the study and praised it with almost no requested revisions, we added the standard deviation (SD) values of the creatinine measurements to the graph as requested. Accordingly, the mean SD values were incorporated in Figure 1. Once again, we sincerely thank the reviewer for his very positive assessment.
Round 2
Reviewer 1 Report
Comments and Suggestions for Authors
Dear Authors,
Thank you for submitting your revised manuscript and rebuttal.
However, the manuscript still requires Major Revision as two critical, unaddressed issues persist:
- Selection Bias (RN vs. PN): The study cohort limited exclusively to patients who underwent Radical Nephrectomy (RN). This inherent selection bias, omitting patients treated with Partial Nephrectomy (PN) for cT1b tumors, significantly limits the generalizability of your conclusion regarding the true incidence of aggressive pathological features. This methodological flaw must be addressed.
- Unjustified Aggressive Feature: The inclusion of "multiple tumors" as an aggressive pathological feature in your definition still lacks clear, evidence-based justification in the revised manuscript or rebuttal. The rationale for this classification must be substantiated with literature or supporting data, or this criterion must be removed.
These two fundamental concerns must be fully rectified. The manuscript remains a Major Revision.
Sincerely,
Reviewer
Author Response
Hello,
We fully acknowledge the reviewer’s concern regarding selection bias and the absence of a Partial Nephrectomy (PN) comparator group. It is important to clarify that our study was not designed as a comparative analysis between PN and RN. Instead, it is a descriptive observational cohort concentrating solely on patients with T1b RCC and moderate to high RENAL scores (≥8) who underwent radical nephrectomy.
This design was intentional for several reasons:
-
In our institution, most patients presenting with T1b RCC and RENAL scores ≥8 during the study period were managed with RN, largely due to anatomical complexity, hilar involvement, central/endophytic tumor location, and surgeon consensus.
Our primary aim was to characterize postoperative renal function, perioperative outcomes, and—importantly—the true incidence of aggressive pathological features within this specific, high-complexity cohort. - Why PN patients are not included:
Introducing a PN cohort would require a separate selection pathway, introduce non-equivalent anatomy, and risk further bias because PN was not routinely offered for RENAL ≥8 tumors during the study period. Including a small, highly selected PN subgroup would itself exacerbate selection bias. -
To our knowledge, no existing study has simultaneously reported:
* T1b tumors, RENAL ≥8 complexity, medium-term renal function (30 months), detailed pathological aggressiveness profiling.
-
Thus, the cohort remains clinically meaningful, even without PN comparison.
-
We have added a clear paragraph in the Discussion (green bolded) section explicitly addressing this point and clarifying the scope of the study.
5. Inclusion of “Multiple Tumors” as an Aggressive Pathological Feature:
Firstly, thank you for highlighting this important conceptual issue. We agree that further justification was needed.
We have added a paragraph in the discussion session (green and bolded) dealing with multifocality and included references.
Sincerely,
Dr. Asali